# Development and Validation of a Rapid Screening Test for HTLV-I IgG Antibodies

**DOI:** 10.3390/v15010129

**Published:** 2022-12-31

**Authors:** Bobby Brooke Herrera, Rafaela Mayoral, Carlos Brites

**Affiliations:** 1Division of Allergy, Immunology, Infectious Diseases, Department of Medicine, Child Health Institute of New Jersey, Rutgers Robert Wood Johnson Medical School, Rutgers University, New Brunswick, NJ 08901, USA; 2Rutgers Global Health Institute, Rutgers University, New Brunswick, NJ 08901, USA; 3Department of Immunology and Infectious Diseases, Harvard T.H. Chan School of Public Health, Boston, MA 02115, USA; 4Universidade Federal da Bahia, Salvador 40170-110, Brazil

**Keywords:** HTLV, diagnosis, screening, testing, rapid antibody test

## Abstract

Initial diagnosis of human T cell lymphotropic virus (HTLV) infections is mainly based by detecting antibodies in plasma or serum using laboratory-based methods. The aim of this study was to develop and evaluate a rapid screening test for HTLV-I antibodies. Our rapid screening test uses HTLV-I p24 antigen conjugated to gold nanoparticles and an anti-human IgG antibody immobilized to a nitrocellulose strip to detect human HTLV-I p24-specific IgG antibodies via immunochromatography. Performance of the rapid screening test for HTLV-I was conducted on a total of 118 serum specimens collected in Salvador, Bahia, the epicenter for HTLV-1 infection in Brazil. Using a Western blot test as the comparator, 55 serum specimens were HTLV-I positive, 5 were HTLV-I and HTLV-II positive, and 58 were negative. The sensitivity of the rapid screening test for HTLV-1 was 96.7% and the specificity was 100%. The rapid screening test did not show cross-reaction with serum specimens from individuals with potentially interfering infections including those caused by HTLV-II, HIV-I, HIV-II, hepatitis A virus, hepatitis B virus, hepatitis C virus, herpes simplex virus, Epstein–Barr virus, SARS-CoV-2, *Chlamydia trachomatis*, *Neisseria gonorrhoeae*, *Treponema pallidum*, *Toxoplasma gondii*, and *Plasmodium falciparum*. The rapid screening test also did not show cross-reaction with potentially interfering substances. Strategies for HTLV diagnosis in non- and high-endemic areas can be improved with low-cost, rapid screening tests.

## 1. Introduction

Human T cell lymphotropic virus (HTLV) was the first retrovirus to be discovered in humans [1]. While four HTLV viruses (types I to IV) have been identified, only HTLV-I and HTLV-II have been demonstrably linked to human disease [1,2,3,4]. HTLV-I is the causative agent of Adult T cell leukemia (ATL) and HTLV-associated myelopathy/tropical spastic paraparesis (HAM/TSP), and other diseases [5]. HTLV-II generally causes no signs or symptoms; however, some individuals infected by HTLV-II can later develop neurological conditions and/or chronic lung infections [6,7]. HTLV-I and HTLV-II are spread globally, with high HTLV-I endemicity in Japan, Oceania, the Middle East, the Caribbean, and parts of South America (e.g., Brazil) and Africa [8,9,10,11].

HTLV is transmitted primarily via mother-to-child routes, sexual contact, and through contaminated needles shared among drug users [12,13,14]. HTLV infections can also occur through transfusion of infected blood specimens or organ transplantation [15,16]. To mitigate these risks, most developed and several developing countries have implemented protocols to screen blood products for HTLV-I/II [17,18]. However, in many countries where HTLV is considered non-endemic, limited screening occurs.

A number of laboratory-based kits that use synthetic peptides and/or viral lysates are commercially available for initial diagnosis of HTLV infections. These kits use various detection techniques including Western blot (WB), enzyme-linked immunosorbent assay (ELISA), or electrochemiluminescent immunoassay (ECLIA) and CLIA to capture HTLV-I/II antibodies in serum or plasma [19]. However, these kits often have decreased accuracy, especially in low prevalence populations, or fail to differentiate between HTLV-I and HTLV-II because the two virus types share high sequence homology [20,21]. In 2014, the FDA approved the MP Diagnostics HTLV Blot 2.4 test, which uses a combination of recombinant HTLV-I/II proteins and HTLV-I viral lysate. Using the MP Diagnostic HTLV Blot 2.4 test, several studies found many WB-indeterminate results in specific population groups including low-risk blood donors [22,23,24]. To reduce the number of inconclusive WB results, other confirmatory tests have been developed including serological (e.g., INNO-LIA HTLV) or molecular assays (e.g., quantitative polymerase chain reaction, qPCR) with higher accuracy [25,26,27].

There is a need for low-cost, non-laboratory-based methods to detect HTLV infections. The aim of this study was to develop and evaluate a rapid screening test for HTLV-I antibodies. The evaluation included cross-reaction analysis using serum specimens from individuals with potentially interfering infections and other potentially interfering substances. Performance of the rapid screening test for HTLV-I was conducted using well-characterized serum specimens collected from individuals in Salvador, Bahia, the epicenter for HTLV-I infection in Brazil [28].

## 2. Methods

### 2.1. Clinical Samples

A total of 118 serum specimens were collected from individuals in Salvador, Bahia, Brazil and initially characterized by a single person using the MP Diagnostics HTLV Blot 2.4 test (MP Biomedicals Asia Pacific Pte Ltd., 2 Pioneer Pl, Singapore, 627885), according to the manufacturer’s instructions. 55 individuals were confirmed HTLV-I positive, 5 were confirmed HTLV-I and HTLV-II positive, and 58 were confirmed negative. Of the 60 HTLV positive individuals, 36 were asymptomatic and 24 were symptomatic. The symptomatic individuals had mild to moderate neurological disease, and none were wheelchair-restricted.

The primary studies under which the serum samples were collected received ethical clearance from the Maternidade Climério de Oliveira (Universidade Federal da Bahia–UFBA) Institutional Review Board (IRB) (approval number: 4.029.414, 15 May 2020). Formal written consent was obtained from all participants, all excess samples were cryopreserved at −80 °C. Prior to analyses, samples were de-identified, thawed, blinded, then immediately processed and results called by a second person. This study received an exemption determination from the UFBA IRB.

### 2.2. Antigen Conjugation to Nanoparticles

HTLV-I p24 antigen (Abcam, Waltham, MA, USA) was conjugated to gold nanoparticles (Abcam, Waltham, MA, USA) according to the manufacturer’s instructions. Briefly, the antigen was diluted to 0.2 mg/mL in supplied dilution buffer. Next, 12 μL of diluted antigen was mixed with 42 μL reaction buffer. 45 μL of the mix was then used to suspend the lyophilized gold nanoparticles. The antigen-nanoparticle mix was incubated for 15 min at room temperature, followed by the addition of 5 μL of supplied quencher solution to stop the coupling reaction. After adding the quencher solution, 100 μL of 1% Tween-20 (MilliporeSigma, Burlington, MA, USA) in PBS (MilliporeSigma, Burlington, MA, USA) and 50 μL of 50% sucrose (MilliporeSigma, Burlington, MA, USA) in water were added to the gold conjugate prior to use in immunochromatography.

### 2.3. Antibody Application to Nitrocellulose Membranes

Nitrocellulose membrane was cut into strips using a laser cutter (Universal Laser Systems, model VLS2.30; 30 watt) at 30% power and 90% speed. Strips were attached to a wick (MilliporeSigma, Burlington, MA, USA) with adhesive paper (DCN Diagnostics, Carlsbad, CA, USA). For the control area, 0.33 μL of anti-HTLV-I p24 antibody (ThermoFisher, Waltham, MA, USA) at 2 mg/mL was spotted on the control area. The test area on the nitrocellulose was generated by pipetting 0.33 μL of anti-human IgG at 6 mg/mL. Strips were air dried and stored in a desiccator at room temperature before use.

### 2.4. Immunochromatography

Each immunochromatography strip was run in a separate microcentrifuge tube. The rapid test solution contained 40 μL of serum specimen, 10 μL of fetal calf serum, 5 μL of quencher solution, and 20 μL of conjugate gold nanoparticle mix. The strips were allowed to react with the mixture for 20 min and then dried and results were image captured using a mobile phone device.

### 2.5. Cross-Reaction Analysis

Serum samples from HTLV-I, HTLV-II, HIV-I, HIV-II, hepatitis A virus, hepatitis B virus, hepatitis C virus, herpes simplex virus, Epstein–Barr virus, SARS-CoV-2, *Chlamydia trachomatis*, *Neisseria gonorrhoeae*, *Treponema pallidum*, *Toxoplasma gondii*, and *Plasmodium falciparum* infections were purchased (LabCorp, Burlington, NC, USA). 40 μL of each serum specimen was processed according to the immunochromatography section in this manuscript.

### 2.6. Interfering Substances Analysis

10 serum specimens were spiked with either sodium heparin (MilliporeSigma, Burlington, MA, USA), sodium citrate (MilliporeSigma, Burlington, MA, USA), EDTA (MilliporeSigma, Burlington, MA, USA), or ACD solution A (MilliporeSigma, Burlington, MA, USA), and 40 μL of each serum specimen was then processed according to the immunochromatography section in this manuscript.

### 2.7. Reproducibility Analysis

The 10 serum specimens from HTLV-I infections were processed in replicates of 5 according to the immunochromatography section in this manuscript.

### 2.8. Image Analysis

The rapid screening test results were machine-read by ImageJ (NIH) to quantify the HTLV-I p24 IgG signals. The average pixel intensity was quantified at the control, background, and test areas. The background-adjusted IgG signal was then normalized to the background-subtracted control area and expressed as % of control. Plots of signals were generated using Prism (version 9.0.0).

### 2.9. Performance Analysis

The sensitivity, specificity, and positive and negative predictive values were calculated using the MedCalc Software (MedCalc Software Ltd., Ostend, Belgium) and are expressed as percentages. In short, sensitivity (or the true positive rate), is the probability that the rapid screening test for HTLV will be positive when the disease is present. While the specificity (or the true negative rate) is the probability that the rapid screening test will be negative when the disease is not present. Using results from both the MP Diagnostics HTLV Blot 2.4 test as the comparator and the rapid screening test for HTLV, we then calculated the sensitivity as the true positives/(true positives + false negatives) and the specificity as true negatives/(false positives + true negatives). Confidence intervals for sensitivity and specificity are “exact” Clopper-Pearson confidence intervals. Confidence intervals for the predictive values are the standard logit confidence intervals.

## 3. Results

To evaluate the rapid screening test for HTLV-I, we compared the sensitivity and specificity using serum specimens collected from individuals with or without HTLV-I and/or HTLV-II in Salvador, Bahia, Brazil. A total of 118 serum specimens were included in the analysis. Using the MP Diagnostics HTLV Blot 2.4 test as the comparator test, 55 serum specimens were confirmed HTLV-I positive, 5 were confirmed HTLV-I and HTLV-II positive, and 58 were confirmed negative (Table 1). Among the HTLV-I positive individuals, 53 of 55 had antibodies against HTLV p19, p24, p26, p28, p32, p36, p53, GD21, and rpg46-I; whereas one individual only had antibodies against p19, p24, GD21, and rpg46-I and the other individual only had antibodies against p19, GD21, and rpg46-I. All 5 HTLV-I and HTLV-II positive individuals had antibodies against p19, p24, p26, p28, p32, p36, p53, GD21, rpg46-I, and rpg46-II. All 58 negative individuals were non-reactive by the MP Diagnostics HTLV Blot 2.4 test.

The rapid screening test had a sensitivity of 96.7% (58/60) and a specificity of 100.0% (58/58) (Table 2, Figure 1 and Figure 2), with positive and negative predictive values of 100.0% and 98.2%, respectively. The test did not detect IgG antibodies against HTLV-I p24 in the two individuals that tested positive by the MP Diagnostics HTLV Blot 2.4 test for HTLV antibodies against p19, GD21, and rpg46-I or p19, p24, GD21, and rpg46-I (patients 51 and 57) (Figure 1A and Figure 2A). The rapid screening test did not detect IgG antibodies against HTLV-p24 in any of the HTLV-I negative individuals confirmed by the MP Diagnostics HTLV Blot 2.4 test (Figure 1B and Figure 2B). Altogether, these results confirm the potential use of a rapid screening test for HTLV-I.

To determine whether the rapid screening test for HTLV-I cross-reacted with antibodies against potentially interfering pathogens, we tested individual serum specimens from infections caused by HTLV-I, HTLV-II, HIV-I, HIV-II, hepatitis A virus, hepatitis B virus, hepatitis C virus, herpes simplex virus, Epstein–Barr virus, SARS-CoV-2, *Chlamydia trachomatis*, *Neisseria gonorrhoeae*, *Treponema pallidum*, *Toxoplasma gondii*, and *Plasmodium falciparum*. The rapid screening test reacted to 10 individual serum specimens from HTLV-I infections, but the test did not react with any other serum specimens from other infections (Table 3, Figure 3). Next, we tested 40 serum specimens spiked with either sodium heparin, sodium citrate, EDTA, or ACD solution A. The rapid screening test for HTLV-I antibodies, did not react with any of the potentially interfering substances (Table 3, Figure 3). The 10 serum specimens from HTLV-I infections were then tested for reproducibility, whereby each specimen was run in replicates of 5. The overall reproducibility of the rapid screening test for HTLV-I was 100% (50/50) (Table 4).

## 4. Discussion

HTLV-I seroprevalence data has largely been based on known endemic regions, with a scarcity of reliable estimates from highly populated countries. The most recent global estimates for the total number of people living with HTLV-I ranges from 5 to 10 million; however, infection counts are likely undercounted [8]. The distribution of HTLV-I infection continues to be focal, with known areas of high prevalence in Japan, Oceania, the Middle East, the Caribbean, and parts of South America and Africa [29]. To complicate the issue further, laboratory-based methods to screen and diagnose HTLV-I infection have been diverse and change over time, limiting the ability to systematically collect infection and associated disease estimates, especially in resource-limited regions [29]. 

Under current testing strategies, to screen and diagnose HTLV-I infection requires an algorithm involving two to three different laboratory-based assays [19]. Most of the assays used to screen HTLV-I infection detect antibodies in serum or plasma [20,21]. Molecular-based tests have been developed, although are not commercially available, and are used primarily to confirm the presence of HTLV-I nucleic acid sequences [26,27]. Even in high-income settings, there is a lack of accessible testing technologies for HTLV-I. The World Health Organization and the Pan American Health Organization have identified as a priority the need to develop low-cost, non-laboratory-based tests for HTLV-I screening [29,30].

For this study, we developed a rapid screening test for HTLV-I antibodies. The test uses HTLV-I p24 antigen conjugated to gold nanoparticles and an anti-human IgG antibody immobilized to a nitrocellulose strip to detect human HTLV-I p24-specific IgG antibodies via immunochromatography (Figure 4). The current cost to produce the test is $5 USD, although with scaled manufacturing the price should decrease significantly. In general, a manufacturing cost of public health screening tests should ideally be near $1 USD.

To analytically validate the rapid screening test, we conducted cross-reaction studies with potentially interfering infections. The test did not show cross-reaction with potentially interfering infections including those caused by HTLV-II, HIV-I, HIV-II, hepatitis A virus, hepatitis B virus, hepatitis C virus, herpes simplex virus, Epstein–Barr virus, SARS-CoV-2, *Chlamydia trachomatis*, *Neisseria gonorrhoeae*, *Treponema pallidum*, *Toxoplasma gondii*, and *Plasmodium falciparum*. We then conducted additional cross-reaction studies with potentially interfering substances including sodium heparin, sodium citrate, EDTA, or ACD solution A. Similarly, the test did not show any cross-reaction with serum spiked with the substances. To determine reproducibility, we ran 10 independent serum specimens from HTLV-I infections in replicates of 5. The rapid screening test reacted 100% of the time.

Next, we performed a clinical evaluation of the rapid screening test in serum specimens collected from individuals in Salvador, Bahia, Brazil. Using the FDA-approved MP Diagnostics HTLV Blot 2.4 test as the comparator test, 55 serum specimens were confirmed HTLV-I positive, 5 were confirmed HTLV-I and HTLV-II positive, and 58 were confirmed negative. When the serum specimens were processed via the rapid screening test, the sensitivity and specificity of the assay was 96.7% and 100%, respectively. The rapid test did not react in serum specimens in two HTLV-I positive individuals; one of the individuals was not HTLV p24 positive by the MP Diagnostics HTLB Blot 2.4 test, however, the other individual did have p24 antibodies. Our analytical and clinical results demonstrate high sensitivity and specificity, potentially enabling lower-cost, more efficient HTLV-I screening.

There were several limitations to this study. Future work should broaden the evaluation to additional settings, sample types, and disease states. While we used serum as the primary sample type, a point-of-care test for HTLV should be optimized for use with fingerstick whole blood, which will further simplify the screening process. Our analysis included 36 asymptomatic and 24 symptomatic HTLV-infected individuals. Regardless of disease state, the rapid screening test performed with high sensitivity and specificity. However, testing should be expanded on HTLV-infected individuals with HAM/TSP. There were two HTLV-I positive samples that were negative by the rapid screening test. These samples demonstrated seroconversion by the comparator assay, where one sample had HTLV-I p24 antibodies, but the other did not. Performance testing on prospectively collected samples in larger cohorts will further corroborate preliminary findings; additionally, evaluations should include testing using seroconverter specimens. Finally, development of a rapid screening test for HTLV-II antibodies should be considered. While the rapid screening test did not cross-react with serum specimens from HTLV-II infections in our analytical studies, more robust testing is warranted.

Without a vaccine and with limited available treatment options for HTLV-related diseases, it is critical to be able to easily identify those who are infected in both endemic and non-endemic regions. While current screening and diagnostic methods for HTLV infection have been shown to be relatively effective, there are no tests that can be performed outside of a laboratory, challenging widescale surveillance and seroprevalence estimates. Screening of HTLV infection, therefore, requires inexpensive, rapid, and accurate non-laboratory-based technology that can be scaled and performed globally. Immunochromatographic, serological-based tests fit these needs.

## Figures and Tables

**Figure 1 viruses-15-00129-f001:**
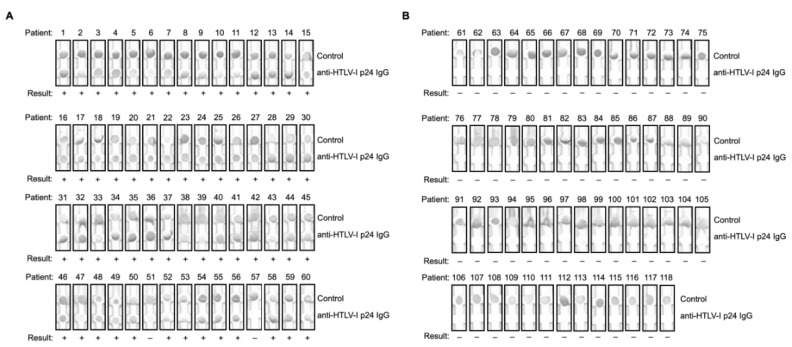
Screening rapid test performance on HTLV-I positive, HTLV-I and HTLV-II positive, or negative serum specimens collected in Salvador, Bahia, Brazil. Serum specimens collected from (**A**) HTLV-I or HTLV-I and HTLV-II infected individuals (*n* = 60) or (**B**) HTLV negative individuals (*n* = 58) were processed by the rapid screening test, then image captured.

**Figure 2 viruses-15-00129-f002:**
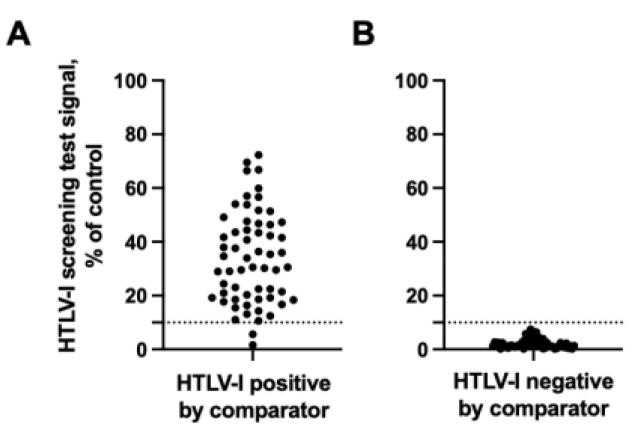
Image analysis of the rapid screening test for HTLV-I. The HTLV p24-specific IgG signal from the rapid screening test was determined and plotted as a % of control for individuals who tested (**A**) positive or (**B**) negative by the comparator test. The y-axis corresponds to the background subtracted IgG signal normalized to the control for each test. Dashed line, cutoff.

**Figure 3 viruses-15-00129-f003:**
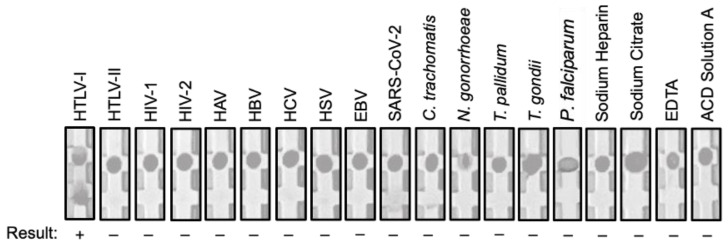
Representative image of the rapid screening test in cross-reaction studies with potentially interfering infections and substances. Serum specimens collected from infected individuals or spiked with substances were processed by the rapid screening test for HTLV-I then image captured. HAV, hepatitis A virus; HBV, hepatitis B virus; HCV, hepatitis C virus; HSC, herpes simplex virus; EBV, Epstein–Barr virus.

**Figure 4 viruses-15-00129-f004:**
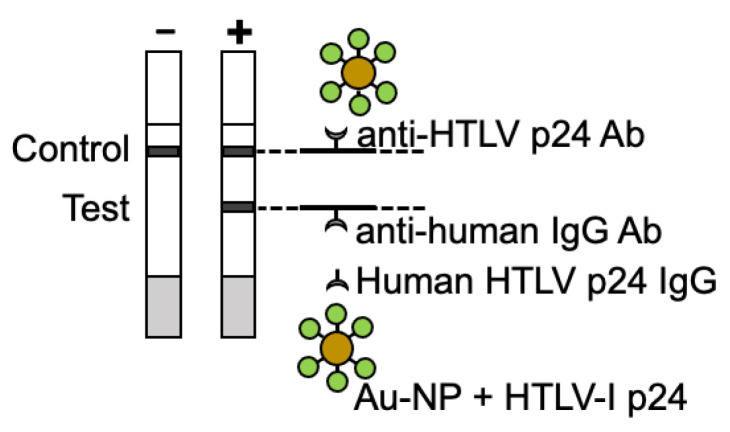
Schematic of the rapid screening test for HTLV-I. The test uses HTLV-I p24 antigen conjugated to gold nanoparticles and an anti-human IgG antibody immobilized to a nitrocellulose strip at the “Test” area to detect human HTLV-I p24-specific IgG antibodies via immunochromatography. An HTLV-I p24-specific antibody is also immobilized to the nitrocellulose strip at the “Control” area to assess flow of the patient sample and manufacturing of the product.

**Table 1 viruses-15-00129-t001:** MP Diagnostics Blot 2.4 Test and Rapid Screening Test for HTLV-I test results from serum specimens collected in Salvador, Bahia, Brazil.

Infection(s)	Number	MP Diagnostics Blot 2.4 Test				Rapid Screening Test
		p19, p24, p26, p28, p32, p36, p53, GD21, and rpg46-I	p19, p24, GD21, and rpg46-I	p19, GD21, and rpg46I	p19, p24, p26, p28, p32, p36, p53, GD21, rpg46-I, and rpg46-II	p24
HTLV-I	55	53/55 (96.4%)	1/55 (1.8%)	1/55 (1.8%)	-	53/55 (96.4)
HTLV-I/HTLV-II	5	-	-	-	5/5 (100.0%)	5/5 (100.0%)
Negative	58	-	-	-	-	0/58 (0.0%)

**Table 2 viruses-15-00129-t002:** Performance statistics of Rapid Screening Test for HTLV-I.

Statistic	Value	95% Confidence Interval
Sensitivity	96.4%	87.5% to 99.6%
Specificity	100.0%	93.8% to 100.0%
Positive Predictive Value	100.0%	-
Negative Predictive Value	98.2%	88.2% to 99.1%

**Table 3 viruses-15-00129-t003:** Cross-reaction studies with potentially interfering infections and substances.

Potentially Interfering Infections	Reactive	Non-Reactive
HTLV-I	10/10	0/10
HTLV-II	0/10	10/10
HIV-1	0/10	10/10
HIV-2	0/10	10/10
Hepatitis A Virus	0/10	10/10
Hepatitis B Virus	0/10	10/10
Hepatitis C Virus	0/10	10/10
Herpes Simplex Virus	0/10	10/10
Epstein–Barr Virus	0/10	10/10
SARS-CoV-2 (COVID-19)	0/10	10/10
*C. trachomatis* (chlamydia)	0/10	10/10
*N. gonorrhoeae* (gonorrhea)	0/10	10/10
*T. pallidum* (syphilis)	0/10	10/10
*T. gondii* (toxoplasmosis)	0/10	10/10
*P. falciparum* (malaria)	0/10	10/10
**Potentially** **Interfering Substances**	**Reactive**	**Non-Reactive**
Sodium Heparin	0/10	10/10
Sodium Citrate	0/10	10/10
EDTA	0/10	10/10
ACD Solution A	0/10	10/10

**Table 4 viruses-15-00129-t004:** Reproducibility studies with HTLV-I serum specimens.

HTLV-1 Reproducibility	Reactive	Non-Reactive
Serum Specimen 1	5/5	0/5
Serum Specimen 2	5/5	0/5
Serum Specimen 3	5/5	0/5
Serum Specimen 4	5/5	0/5
Serum Specimen 5	5/5	0/5
Serum Specimen 6	5/5	0/5
Serum Specimen 7	5/5	0/5
Serum Specimen 7	5/5	0/5
Serum Specimen 9	5/5	0/5
Serum Specimen 10	5/5	0/5

## Data Availability

All data produced in the present study are available upon reasonable request to the corresponding author.

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
