# Peer review of "Development and Validation of a Rapid Screening Test for HTLV-I IgG Antibodies"

_viruses, 2022, doi:10.3390/v15010129_

Round 1

Reviewer 1 Report

This manuscript reports a new screening test for HTLV antibody. Usually, the characteristics of screening tests should be easy to operate, good sensitivity and specificity, in which sensitivity is particularly important. The confirmation method is often complicated operation process, low sensitivity, but good specificity. Therefore, it is suggested that the author supplement the results compared with commercial screening test, and use the results of confirmation method as negative and positive results to evaluate the specificity and sensitivity of reagents.

Author Response

Response to Reviewer 1:

  • This manuscript reports a new screening test for HTLV antibody. Usually, the characteristics of screening tests should be easy to operate, good sensitivity and specificity, in which sensitivity is particularly important. The confirmation method is often complicated operation process, low sensitivity, but good specificity. Therefore, it is suggested that the author supplement the results compared with commercial screening test, and use the results of confirmation method as negative and positive results to evaluate the specificity and sensitivity of reagents.

Response: We agree that a screening test should be easy to operate with robust accuracy, including both sensitivity and specificity. We also agree that confirmatory tests can often be complex, requiring laboratory-intensive processes. In the case of our study, we compared the our new rapid screening test to the FDA-approved MP Diagnostics HTLV Blot 2.4, used as a screening test and also confirmatory test depending on the setting. We then calculated the sensitivity and specificity of our test compared to the comparator (MP Diagnostics HTLV Blot 2.4). The sensitivity of the new rapid screening test was 96.4% and the specificity was 100%. These results are detailed in Table 3 and Table 4 of the manuscript.

Reviewer 2 Report

The paper is well written and considered. I agree that there is a clear need for inexpensive, sensitive and efficient HTLV testing strategies. This methodology seems to suggest an effective solution and shows successfully that the test is selective for HTLV-1 over several other interfering agents or other common viruses. However, there is some ambiguity for me with the test results for sample 51 and 57, which calls into question some of the reported confidence of the test, although overall it seems sensitive and selective on this small number of samples.

I wonder if the authors could make some small revisions and clarify a few points:

  1. Please detail in the methodology how the serum was stored? The current methodology just says the serum was “banked” - please detail storage conditions.

  2. Please detail in the methodology how the testing result was confirmed? Was it by one person or checked and confirmed positive or negative by one person initially and then by a second person, for example? Were the individuals recording the result blinded to the sample status in the MP test?

  3. References 15, 27, 28 & 29 appear to be missing a DOI, please add the DOI if they are available?

More substantial suggestion:

To tie in with point 2. Were computerized confirmatory testing methods considered? Whilst this would not be useful if the test were developed for the public at least in this development stage of assessing whether the test should move forward, having more certainty in it would certainly be useful. In the absence of at the very least double confirmatory result testing by two independent non-biased individuals, something like ImageJ could be used to assess positive or negative (and potentially set cut-offs?) results using a computer rather than a person and remove any potential user-bias. 

For me, there is a spot on the sample 51 in my printout and when zoomed into the image in the pdf; all the negative samples in figure 2 are clearly negative. Looking at sample 51 next to a negative it is difficult to ignore a small shaded area of positivity, which could lead to some ambiguity. Sample 57 there is a smear on the sample. Could both these samples be repeated and new images used in the paper or at least commented upon. Could the authors also say why computerized confirmatory testing was not used?

Author Response

Response to Reviewer 2:

  • Please detail in the methodology how the serum was stored? The current methodology just says the serum was “banked” - please detail storage conditions.

Response: Thank you for this suggestion. We have added language to clarify that the samples were cryopreserved at -80°C prior to analyses (lines 81-81). 

  • Please detail in the methodology how the testing result was confirmed? Was it by one person or checked and confirmed positive or negative by one person initially and then by a second person, for example? Were the individuals recording the result blinded to the sample status in the MP test?

Response: We have added language in the methods section to describe how sample processing occurred (lines 71-84). The samples were initially processed by the FDA-approved MP Diagnostics HTLV Blot 2.4 as the comparator by a single person. This person then de-identified and blinded the samples, after which a second person then processed and called the results from the new rapid screening test.

  • References 15, 27, 28 & 29 appear to be missing a DOI, please add the DOI if they are available?

Response: Thank you. I have added DOIs to references 27 and 28; we also double-checked references 15 and 29, which do not have DOIs.

  • To tie in with point 2. Were computerized confirmatory testing methods considered? Whilst this would not be useful if the test were developed for the public at least in this development stage of assessing whether the test should move forward, having more certainty in it would certainly be useful. In the absence of at the very least double confirmatory result testing by two independent non-biased individuals, something like ImageJ could be used to assess positive or negative (and potentially set cut-offs?) results using a computer rather than a person and remove any potential user-bias. 

Response: Initially, computerized testing methods were not used. However, after your suggestion, we decided to perform image analysis on the rapid screening test results. We have published elsewhere on this image analysis methodology including for SARS-CoV-2 antigen (here and here) and recombinase polymerase amplification (here) tests. Using this approach, we computerized the rapid screening test results and have included details about our methodology in the Methods section (lines 134-138) and our plots as a new Figure 2 in the Results. The two individuals (patient 51 and 57) that were negative by the rapid screening test were also negative by image analysis; there were others that were near the cutoff, but remained positive by image analysis. All of the negatives were also negative by image analysis.

  • For me, there is a spot on the sample 51 in my printout and when zoomed into the image in the pdf; all the negative samples in figure 2 are clearly negative. Looking at sample 51 next to a negative it is difficult to ignore a small shaded area of positivity, which could lead to some ambiguity. Sample 57 there is a smear on the sample. Could both these samples be repeated and new images used in the paper or at least commented upon. Could the authors also say why computerized confirmatory testing was not used?

Response: Thank you for your comments. Unfortunately, we are unable to perform repeat testing for patient sample 51. Moreover, it is not uncommon for rapid tests with very minor shadows to be considered negative. Our image analysis also determined that the signal is negative and below the cutoff. Two tests were run for patient 57. While the first test had non-specific smear near the edge of the nitrocellulose paper, the second test also resulted in a clear negative by eye and by image analysis. As such, Figure 1 has been updated in the manuscript.

Reviewer 3 Report

The development of a PoC test for HTLV-1 diagnosis is considered a priority. The authors developed and did a preliminary validation of a rapid test for the detection of IgG to p24 HTLV-1 protein. The assay is promising and the manuscript is well written.

Some comments:

1. More details regarding the interpretation of results is needed. Was it done directly by visual inspection of the strip, or by evaluating the photo? The quality of photo, luminosity may influence the interpretation. Is there an app or reader in the phone to read it? Only one person assessed all results? Same person always for all samples? Was this person blind to the HTLV status? This information needs to be included. In the future, authors should consider assessing concordance between different technitians. 

2. Authors should consider testing for cross-reactivity with samples obtained from patients infected by Plasmodium as earlier HTLV-1 assays reported cross reaction with such samples. 

3. Any information about clinical status for patients included in the study? This is very important as it is known that patients with HTLV-1 associated diseases, particularly HAM, usually have higher antibodies titer, and therefore this would influence the assay sensitivity.

4. Authors should avoid saying that the current available assays have poor sensitivity and specificity. ELISA and CMIA assays currently in use for the screening of HTLV have 100% of sensitivity and 98.1-99.5% specificity (da Silva Brito et al 2018). Therefore, this is not accurate. The necessity of a rapid test is due to the need to increase access to diagnosis and rapid turn-around time. Not with the performance of the screening tests. 

5. I would recommend adding after 2.4 a session explaining the sensitivity and specificity analysis, with description of the samples tested. I would then start the results with this information (150-179) before the data on cross-reactivity, reproducibility and interferences. 

6. Authors should include that the development of PoC test was identified a priority by WHO and PAHO (references to their reports on HTLV). This will strengthen their work.

7. A very important point refers to cost. An accessible PoC test for HTLV must be low-cost. Authors should add information reg cost of the assay.

8. Authors need to expand discussion on the use of plasma. A rapid and mainly important PoC test will greatly benefit of using blood as analyte. 

9. Minor comments: Line 195 Even in high income settings, HTLV-1 testing is not accessible.

Author Response

Response to Reviewer 3:

  • More details regarding the interpretation of results is needed. Was it done directly by visual inspection of the strip, or by evaluating the photo? The quality of photo, luminosity may influence the interpretation. Is there an app or reader in the phone to read it? Only one person assessed all results? Same person always for all samples? Was this person blind to the HTLV status? This information needs to be included. In the future, authors should consider assessing concordance between different technicians. 

Response: Computerized testing methods were not initially used. However, after your suggestion, we decided to perform image analysis on the rapid screening test results. We have published elsewhere on this image analysis methodology including for SARS-CoV-2 antigen (here and here) and recombinase polymerase amplification (here) tests. Using this approach, we computerized the rapid screening test results and have included details about our methodology in the Methods section (lines 134-138) and our plots as a new Figure 2 in the Results. The two individuals (patient 51 and 57) that were negative by the rapid screening test were also negative by image analysis; there were others that were near the cutoff, but remained positive by image analysis. All of the negatives were also negative by image analysis. This is also further explained in the methods section.

  • Authors should consider testing for cross-reactivity with samples obtained from patients infected by Plasmodium as earlier HTLV-1 assays reported cross reaction with such samples. 

Response: Thank you for the suggestion. We included analysis of the rapid test with serum from patients infected with P. falciparum with no cross-reaction.

  • Any information about clinical status for patients included in the study? This is very important as it is known that patients with HTLV-1 associated diseases, particularly HAM, usually have higher antibodies titer, and therefore this would influence the assay sensitivity.

Response: We have added information about clinical status under the ‘Clinical Samples’ of the Methods section (lines 74-77).

  • Authors should avoid saying that the current available assays have poor sensitivity and specificity. ELISA and CMIA assays currently in use for the screening of HTLV have 100% of sensitivity and 98.1-99.5% specificity (da Silva Brito et al 2018). Therefore, this is not accurate. The necessity of a rapid test is due to the need to increase access to diagnosis and rapid turn-around time. Not with the performance of the screening tests. 

Response: Thank you for emphasizing this point – we agree. We have clarified these points throughout the manuscript.

  • I would recommend adding after 2.4 a session explaining the sensitivity and specificity analysis, with description of the samples tested. I would then start the results with this information (150-179) before the data on cross-reactivity, reproducibility and interferences. 

Response: Thank you for this suggestion. We have added to our methods section on the sensitivity and specificity analysis (lines 142-148). As suggested, we have also re-worked the entire results section to begin with the sensitivity/specificity of the test followed by data on cross-reactivity, reproducibility, and interferences.

  • Authors should include that the development of PoC test was identified a priority by WHO and PAHO (references to their reports on HTLV). This will strengthen their work.

Response: Thank you for the suggestion. We have added that PAHO and the WHO have called for the development of point-of-care testing technology for HTLV in the discussion with associated references (lines 242-244).

  • A very important point refers to cost. An accessible PoC test for HTLV must be low-cost. Authors should add information reg cost of the assay.

Response: We strongly agree that a point-of-care screening test for HTLV – and other infectious diseases – should be low-cost. We have added language in the discussion around cost for the test (lines 281-283).

  • Authors need to expand discussion on the use of plasma. A rapid and mainly important PoC test will greatly benefit of using blood as analyte. 

Response: We agree point-of-care screening tests should ideally be used with whole blood as the primary sample type. We have added a statement in our discussion (lines 314-316). 

  • Minor comments: Line 195 Even in high income settings, HTLV-1 testing is not accessible.

Response: Thank you for your suggestion. We have adjusted the wording in this sentence.

Round 2

Reviewer 2 Report

The authors have responded to all my suggestions and the manuscript is much improved. The methodology is clearer and the suggested computerised (unbiased) analysis gives increased robustness and confidence in the test. The new Figure 2 is clear and improves the manuscript. Adding other agents to check the specificity is beneficial. Information on the cost of the test is useful information.

Author Response

Thanks for your suggestions.